



# A mixed distribution approach for low-flow frequency analysis – Part 2: Modeling dependency using a copula-based estimator

Gregor Laaha

University of Natural Resources and Life Sciences, Vienna, Department of Landscape, Spatial and Infrastructure Sciences, Institute of Statistics, Peter Jordan-Straße 82/I, 1190 Vienna, Austria

**Correspondence:** Gregor Laaha (gregor.laaha@boku.ac.at)

**Abstract.** In climates with a warm and a cold season, low-flows are generated by different processes, which violates the homogeneity assumption of extreme value statistics. In this second part of a two-part series, we extend the mixed probability estimator of the companion paper (Laaha, G., 2022) to deal with dependency of seasonal events. We formulate a copula-based estimator for seasonal minima series and examine it in a hydrological context. The estimator is a valid generalization of the annual probability estimator and provides a consistent framework for estimating return periods of summer, winter and annual events. Using archetypal examples we show that differences to the mixed estimator are always observed in the upper part of the distribution, which is less relevant for low-flow frequency analysis. The differences decrease as the return period increases so that both models coincide for the severest events. In a quantitative evaluation, we test the performance of the copula estimator on a pan-European data set. We find a large gain of both mixed distribution approaches over the annual estimator, making these approaches highly relevant for Europe as a whole. We then examine in more detail the relative performance gain of the mixed copula versus the mixed distribution approach. The analysis shows that the gain for the 100-year event is actually minimal. However, the gain for 2-year events is considerable in some of the catchments, with a relative deviation of -15 to -23 % in the most affected regions. This points to a prediction bias of the mixed probability estimator that can be corrected using the copula approach. Using multiple regression models, we show that the performance gain can be well explained on hydrological grounds, with weak seasonality leading to a high potential for corrections and strong seasonal correlation reinforcing the need to take it into account. Accordingly, the greatest differences can be observed in mid-mountain regions in cold and temperate climates, where rivers have a strongly mixed low-flow regime. This finding is of particular relevance for event mapping, where regional severity can be misinterpreted when the seasonal correlation is neglected. We conclude that the two mixed probability estimators are quite similar, and both are more accurate as the annual minima approach. In regions with strong seasonal correlation the mixed copula estimator is most accurate and should be preferred over the mixed distribution approach.

## 1 Introduction

In seasonal climates with a warm and a cold season, low-flows are generated by different processes so that the annual extreme series will be a mixture of summer and winter low-flow events. This can violate the basic assumption of extreme value statistics and give rise to inaccurate conclusions. In the first part of the study (Laaha, G., 2022) we addressed the problem of process het-





erogeneity by proposing a mixed probability estimator that combines the seasonal distributions. Compared to the conventional estimator, we showed that the estimator can estimate return periods indeed more accurately.

In this second part of the study, we address the problem when summer and winter events are not completely independent. This can play a role in low-flow statistics, where the events have a long time scale and may last for several month, or even some year (Stahl and Hisdal, 2004). Although not all events span such a long time period, there are instances where summer

low-flows are continued by a winter low-flow event. Van Loon et al. (2015) found this drought type quite common in cold temperate climates, whereas the reverse case of a prolonged winter drought appeared to be less frequent. Prolonged low-flow events need not be extreme in both seasons, but some dependence between seasonal events must still be expected. For such cases, the mixed probability model can be extended by using a multivariate distribution approach.

The use of multivariate distributions for extremes dates back to Gumbel (1960) and has a long tradition in hydrology. A well-

known problem in flood frequency analysis is that the extreme event cannot be fully described by the peak of the event alone, but additional aspects such as duration and volume must be considered. Yue et al. (1999) employed a bivariate Gumbel distribution model to perform a consistent frequency analysis of correlated pairs of variables, such as peak and volume, and volume and duration. The model was found to be well suited to represent the joint occurrence probability of flood characteristics. The approach uses an analytical formulation of the multivariate distribution model, which, however, exists only for some

distribution families. In the cases where no analytical expression exists, the joint probability model can be constructed using a copula approach.

Copula models have recently received increasing attention in hydrology. Most examples are for flood frequency analysis but also some examples for low-flows exist. A good overview of copula theory with respect to hydrological applications is given in Klein et al. (2011). Renard and Lang (2007) demonstrated the value of Gaussian copula models for a variety of problems where

two or more variables are required for inference. Examples include field significance in multiple testing problems (e.g. regional trend analysis), regional risk assessment that combines at-site risks at the regional scale, joint assessment of discharge-duration-frequency of design events, and incorporation of intersite-dependence in regional flood frequency analysis. The paper showed the usefulness of the multivariate approach but underlined the limitation of being restricted to a Gaussian copula model. Poulin et al. (2007) found that extreme event characteristics have a particular tail-dependence for which the Gumbel and survival

Clayton copula models should be most adequate. Fischer and Schumann (2021) again used Vine copulas to determine the most probable combinations of flood-types for tributaries. Yet another example is given in Klein et al. (2011), which showed the potential of copulas for dam safety analysis. They used a bivariate copula distribution model to define joint return periods of critical flood peak an volume events, which allowed them to better assess the effect of flood protection measures for dam safety based on their marginal effects.

In a low-flow hydrologic context, bivariate analysis of volume and duration of below-threshold events were performed by Ashkar et al. (1998) using a bivariate exponential distribution. They found the applicability of the method depending on the strength of correlation of the two variables. Sahoo et al. (2020) used a bivariate copula distribution model to analyse cumulative annual drought duration of volume for two stations in an Indian study area. They found the general extreme value distribution (GEV) in combination with a Clayton and Frank copula model best suited in both stations, however the differences between





the copula models appeared to be small and depending on the criterion used for model selection. This view is supported by Genest and Favre (2007) who found the difference between different extreme value copula models to be mostly negligible, and choosing between them would make any serious difference for prediction purposes. Ahn and Palmer (2016) used a non-stationary copula to predict future bivariate low-flow frequency in the Connecticut river basin. In the absence of stationarity, the model was shown to provide more accurate frequency estimates than the stationary model alternatives. A similar approach was used by Jiang et al. (2015) to perform a bivariate frequency analysis for the low-flow series from two neighbouring hydrological gauges. The study extended the scope of the regional model by Ahmadi et al. (2018), which used a stationary copula model to define the dependency structure of low-flow magnitude of tributaries.

While copula approaches have been used either to combine different event characteristics (e.g. volume and duration) in a local analysis or for the same event characteristic at multiple sites, we are not aware of any study that has used copulas to consider seasonal dependence in a mixed distribution approach. In this second paper, we propose such a mixed copula estimator for low-flow frequency analysis in mixed seasonal regimes. The aims of the paper are:

– To formulate an enhanced mixed distribution approach for minima that incorporates seasonal correlation;

– To review its behaviour and plausibility for archetypal catchments;

– To evaluate its possible performance gain compared to the mixed probability estimator;

– To give a recommendation which estimator to use under specific hydrological conditions.

The paper will also extend the analysis of the companion paper (Laaha, G., 2022) to the pan-European level, to assess the value of mixed distribution approaches for a greater variety of hydrologic regimes.

## 2   A mixed copula approach for low-flows

### 2.1   Theoretical probability estimator

Let us consider the case that the annual minima series is composed of events that arise from different processes occurring in the summer and winter season. Here, the annual minima series $AM$ can be viewed as the minima of the annual summer minima $AM_S$ and winter minima $AM_W$:

$$AM = min\{AM_S, AM_W\} \tag{1}$$

These constitute samples from two random variables, annual summer low-flow $S$ and annual winter low-flow $W$, respectively. Assuming independence of summer and winter events, the probability of a low-flow event with magnitude $q$ can be obtained from its occurrence probability in the summer season $F_S(q)$ and winter season $F_W(q)$ using the multiplication rule of statistics. In this case, the mixed probability estimator

$$F_{mix}(q) = 1 - \{1 - F_S(q)\}\{1 - F_W(q)\}, \tag{2}$$





is a valid generalization of the annual frequency estimator that yields improved probability estimates in case of differing

summer and winter distributions (Laaha, G., 2022).

The multiplication rule is just a special case of a more general problem of frequency analysis where we are concerned with finding the joint occurrence probability of events in two variables. This can be solved by bivariate frequency analysis. Here, a bivariate probability model is used to estimate the joint probability that $S$ and $W$ are less or equal than the magnitudes $s$ and $w$, respectively:

$$P(S \leq s, W \leq w) = F_{S,W}(s,w) \tag{3}$$

In this paper we propose a copula-based approach to model the bivariate frequency distribution $F_{S,W}(s,w)$. A copula $C(\cdot)$ is a function which exactly describes and models the dependency structure between correlated random variables independently of their marginal distributions (Klein et al., 2011). The link between the copula and the multivariate distribution is provided by the theorem of Sklar (1959), so that the joint cdf can be written as:

$$F_{S,W}(s,w) = C[F_S(s), F_W(w)] \tag{4}$$

As we are modeling extreme values we use a Gumbel–Hougaard copula of the form:

$$C = \exp\left[ -\left( (-\log(u))^\theta + (-\log(v))^\theta \right)^{1/\theta} \right], \theta \in [1, \infty) \tag{5}$$

where $u$ and $v$ denote the so-called pseudo-observations corresponding to the empirical probabilities of $S$ and $W$, respectively. The parameter $\theta$ is used to model the strength of the dependency structure of the distributions. It is directly linked to rank

correlation (i.e., Spearman's $Rho$), with $\theta = 1$ denoting the case of complete independence.

The model can now be used to estimate the occurrence probability of a low-flow event by inserting its magnitude $q$ into both marginal distributions. This yields to its summer and winter occurrence probabilities $F_S(q)$ and $F_W(q)$, respectively. The (joint) probability that $S$ and $W$ both fall below $q$ is given by the so-called AND operator (Klein et al., 2011, Eq. 8.29 for maxima). Applying the AND operator to minima yields the copula-based mixed probability estimator of annual low-flow

occurrences:

$$F_{mix,C}(q) = F_S(q) + F_W(q) - C[F_S(q), F_W(q)]. \tag{6}$$

As stated above, the Gumbel-Hougaard copula includes the case of independent events, where $C[F_S(q), F_W(q)]$ becomes $F_S(q) \cdot F_W(q)$, and Eq. 6 simplifies to Eq. 2 accordingly. The mixed copula estimator is therefore a valid generalization of the mixed distribution estimator proposed in the companion paper (Laaha, G., 2022) for the case that summer and winter low-

flows are not completely independent. As in the companion paper, we employ a Weibull-distribution, with marginal summer distribution

$$G_S(q) = 1 - exp\left[ -\left( \frac{q - \zeta_S}{\beta_S} \right)^{\delta_S} \right], \tag{7}$$





and marginal winter distribution

$$G_W(q) = 1 - exp\left[-\left(\frac{q - \zeta_W}{\beta_W}\right)^{\delta_W}\right].\tag{8}$$

to model the distributions of seasonally separable events. The $\zeta.$, $\beta.$ and $\delta.$ are the location, scale and shape parameters of summer (index $S$) and winter (index $W$) distributions, respectively.

## 2.2    Empirical probability estimator

In the same way as for the theoretical estimator, we can also define an empirical estimator that generalizes the mixed probability approach to the case of seasonal correlation.

Let $q$ be an element of an annual minima series $AM$ with $n$ observations, and $m$ its rank in increasing order. Assuming that the $AM$ is sampled from (a sequence of) independent and identically distributed random variables, thus *iid*, the empirical probability $p_m(q)$ can be calculated by

$$p_m(q) = \frac{m}{n+1}.\tag{9}$$

The iid assumption implies that the annual minima are generated by the same low-flow process (i.e., are homogeneous) and are
not correlated. In the case where the sample is heterogeneous and possibly correlated, a generalized estimator can be obtained from bivariate frequency analysis. Assuming seasonally separable low-flow distributions, the generalized empirical probability can be written as

$$p_{mix,C}(q) = \frac{m_S}{n_S + 1} + \frac{m_W}{n_W + 1} - C_n\left(\frac{m_S}{n_S + 1}, \frac{m_W}{n_W + 1}\right)\tag{10}$$

where $m_S$ is the rank of an event with magnitude $q$ in the $AM_S$ series and $m_W$ its rank in the $AM_W$ series, respectively.
The $C_n(\cdot)$ is the empirical copula, which defines the empirical multivariate distribution in analogy to Eq. 4. In contrast to the theoretical copula, no parametric model has to be assumed to calculate the empirical probability. The estimator can be rewritten as

$$p_{mix,C}(q) = p_{m,S}(q) + p_{m,W}(q) - C_n\left[p_{m,S}(q), p_{m,W}(q)\right],\tag{11}$$

to conform to the theoretical probability estimator in Eq. 6.
Like any empirical probability estimator, the $p_{mix,C}(q)$ is susceptible to sampling uncertainties and therefore mainly used in distribution plots to visualize the fit of the theoretical distribution to observations. In contrast to the common empirical estimator, the magnitude $q$ of the annual series may be missing from the marginal series, and its empirical probability must be approximated. Such an approximation is obtained by averaging the rank of the next smaller and bigger elements to $q$ in the marginal series, in accordance with Laaha, G. (2022).


## 2.3 Demonstration of model behaviour

As in the companion paper we use archetypal examples to demonstrate the behaviour of the model. The examples used now differ not only in terms of the strength of seasonality, but also in terms of seasonal correlation, ranging from insignificant correlations to highly correlated summer and winter events. An overview is given in Table 1.

The first example is gauge Ebensee at river Langbathbach representing the case of almost no seasonality ($SR$=1.09) and insignificant seasonal correlation, as indicated by a $Rho$ of 0.22 and a p-value of 0.20. The gauge was also used in the companion paper, as it represents a typical foreland catchment, where discharges from the mountains and lowlands mix. The mixed distribution approach has been shown to differ strongly from the annual distribution fitted to the heterogeneous series in this case. Since the assumption of independence of seasonal low-flows cannot be rejected, it is reasonable to assume that there is not much difference between the mixed distribution approach and the copula-based estimator. Figure 1a shows that there is indeed no difference between the two estimators in the lower part of the distribution, which is relevant for low-flow frequency analysis. There are differences in the upper part of the distribution, however, where the mixed distribution falls much below the annual estimator. The mixed copula approach corrects a large part of this deviation. It brings the mixed distribution close to the annual one, which has previously been shown to be appropriate in the central part or the distribution where the summer and winter distributions coincide.

The second example is gauge Weg at river Isen situated in low-hilly terrain in Bavaria, Germany, representing the case of weak summer seasonality ($SR$=0.87) combined with moderate seasonal correlation ($Rho$=0.44). The gauge was also used in the companion paper to show that substantial mixing can occur even in cases where one seasonal distribution is always lower than the other, but in this case not in the most extreme low-flow events. It was shown that mixed probability estimator reflects this behavior by following the dominant distribution at the lower tail and combining the probabilities in the range of discharges where low-flows in both seasons occur. Figure 1b shows that despite the now significant correlation (p-value < 0.01), both mixed probability approaches coincide at the lower part of the distribution. As in the gauge Ebensee example, there is some deviation between the mixed and the annual distribution at higher low-flows, which is again reduced by the mixed copula estimator. We note anew that this behaviour is only observed at the upper part of the distribution, which is less relevant for low-flow statistics. At most, there is a slight difference for moderate low-flow events with a return period of $T$=2 years, but this should have little effect on the estimated return periods for the given case.

The third example is gauge Trausdorf an der Wulka at river Wulka situated in the eastern foreland of the Alps in Burgenland, Austria. It represents the case of a medium sized lowland catchment ($AREA$=236 km$^2$) with significant tributaries from the Alps. Low-flows in the catchment have a strong seasonal correlation ($Rho$=0.71) and moderate summer seasonality ($SR$=0.74). Because of the strong correlation we can expect there is a large difference between the mixed distribution approach and the copula-based estimator. This is indeed visible from Fig. 1c. There is still a part where both mixed distribution approaches coincide, but this is much smaller and restricted to return periods of $T$=10 years and more. For lower return periods, the mixed distribution falls far below the dominant summer distribution, and this difference is to a large extent corrected by the mixed copula estimator.





The forth and final example is gauge Schalklhof at river Schalklbach representing an medium sized alpine catchment
($AREA$=108 km$^2$) in Tyrol, Austria. The catchment represents the ultimate case of a significant seasonal correlation ($Rho$=0.27
with a p-value < 0.05) combined with a very strong winter seasonality ($SR$=2.52). In this case there is almost no mixture of
summer and winter discharges making it irrelevant whether they are correlated or not. Figure 1d shows that all distributions
apart from the subordinate summer distribution coincide, so there is no gain of the mixed copula estimator over the mixed
probability estimator. Moreover, there is also no gain of the mixed distribution approaches over the annual estimator at all, as
the annual distribution and the winter distribution coincide. In the companion paper it was noted that such cases are rare in the
Austrian study area and limited to high-alpine catchments. We expect them to play a subordinate role in the European data set
as well.

On the whole, the examples suggest that there is little difference between the two mixed estimators in the lower part of the
distribution, so using the copula-based approach should have little effect on the estimated return period of an event. Seasonal
correlation is observed to affect mainly the upper part of the distribution, which is less relevant for low-flow analysis. Although
the differences between the two mixed probability estimators tend to be small, they increase with increasing correlation and
may become relevant for highly correlated cases. In the following, we assess the possible gains of the mixed copula estimator
based on a comprehensive pan-European data set.

**Table 1.** Overview of the example gauges used for model demonstration. Shown are the station identifier ($ID$), catchment area ($AREA$),
Spearman's correlation between summer and winter event series ($Rho$) together with its significance level (p-value), the mean resultant of
the circular seasonality index ($r$), and the seasonality ratio ($SR$) representing the ratio between mean summer and mean winter low-flow.

| ID | Name of gauge / river | AREA in km$^2$ | Rho | p-value of Rho | r | SR |
|---|---|---|---|---|---|---|
| 205229 | Ebensee / Langbathbach | 40.0 | 0.22 | 0.20 | 0.49 | 1.09 |
| 18381500 | Weg / Isen | 59.7 | 0.44 | < 0.01 | 0.31 | 0.87 |
| 210328 | Trausdorf an der Wulka / Wulka | 235.9 | 0.71 | < 0.01 | 0.50 | 0.74 |
| 201160 | Schalklhof / Schalklbach | 107.8 | 0.27 | < 0.05 | 0.92 | 2.52 |

## 3 Evaluation using streamflow data

### 3.1 Pan-European assessment


We evaluate the model using streamflow data from the European Water Archive (EWA), which has been used in a number
of regional low-flow studies (e.g. Stahl et al., 2010). The EWA data set consists of long-term daily streamflow data and basic
catchment information across Europe. We use the data selection of Laaha et al. (2017) and Stahl et al. (2020) consisting of
about 800 gauges across Europe with daily measurements for the period 1976-2010 and 2015. The selected stations are free
from major disturbances of the low-flow regime. Further details on this study, including data and methods, are given in Laaha
et al. (2017).



**Figure 1.** Probability plots of summer (red), winter (blue), annual (black), mixed (green) and mixed copula (magenta) distributions for archetypal cases: (a) gauge Ebensee at river Langbathbach situated in the northern foothills of the Alps of Upper Austria, (b) gauge Weg at river Isen situated in low-hilly terrain in Bavaria, Germany, (c) Trausdorf an der Wulka at river Wulka situated in the eastern foreland of the Alps in Burgenland, Austria, and (d) gauge Schalklhof at river Schalklbach representing an alpine catchment in Tyrol, Austria.





The study area covers large parts of western, central and northern Europe and represents a great diversity of hydrological regimes. The diversity is clearly reflected by the low-flow seasonality patterns in Fig. 2, with summer seasonality in the West, and winter seasonality in higher altitudes and in the North and East. The seasonal correlation, which is examined as a key

variable in this paper, shows a different pattern with low values in the Alps and the North and high values in the lowlands and mid-mountain regions in between. The study area is mainly located in a temperate climate, with Oceanic climate in the West, and continental climate in central Europe and the East. Parts of the study area are further located in cold-temperate and sub-arctic climates. These are well represented in the data set by alpine and Norwegian cases. The Mediterranean climate is only represented marginally, through a few catchments in southern France and northern Spain. Due to the mild and humid

winters, the low-flows Mediterranean catchments have a clear summer regime for which the mixed distribution approaches of this work should have little relevance. The study area has both a high gauging density and good data quality and is well suited to assess the effects of seasonal correlation in a variety of seasonal low-flow regimes.

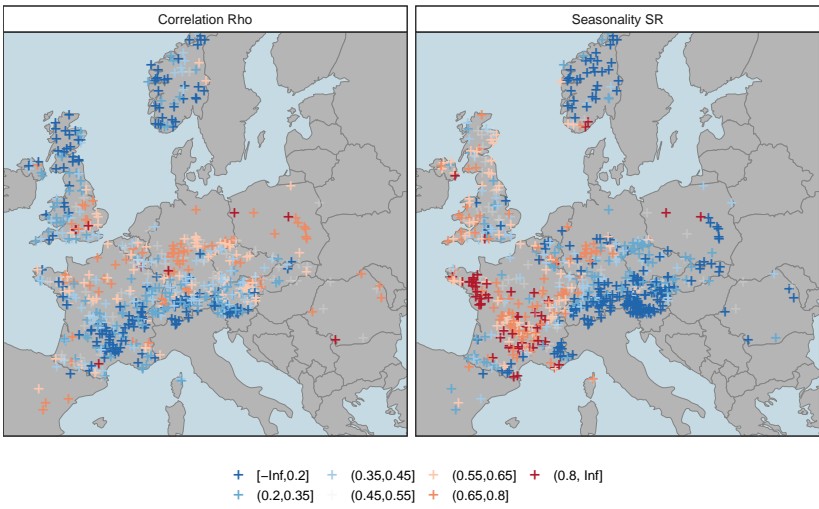

**Figure 2.** Location of gauges and and main features of the study: Seasonal dependency measured by the Spearman's Rho between summer and winter maxima series (left), and the seasonality ratio ($SR$) defined as the ratio of mean winter and mean summer low-flow (right).

### 3.2   Evaluation method

A similar procedure as in the companion paper is used for model evaluation, which is briefly described below. Again, we

assume that the mixed copula estimator is superior to the simpler models, since it is a valid generalization of the annual and the mixed probability estimator (Section 2.1). In this context, the evaluation is carried out in three steps.

In the first analysis, we assess the performance at the continental scale to learn about the relevance of the enhanced estimator in different regions. The evaluation focuses on the change in return period when using a seasonal probability estimator instead of the conventional annual estimator. This change, or *deviation* $d_T$ of a given return period $T$ is obtained by calculating the





associated flow quantile of the annual estimator and inserting it into the alternative probability distribution. This yields the alternative estimates $Tmix$ when using the mixed distribution approach, and $TmixC$ when using the mixed copula estimator. As we are dealing with minima, the return period is calculated as the reciprocal of the occurrence probability, i.e., $T = 1/p$.

Consistent with Laaha, G. (2022), the performance gain of the mixed copula estimator over the conventional annual estimator is assessed based on its relative deviation

$$rd_T = \frac{T - TmixC}{TmixC} \tag{12}$$

and its relative absolute deviation ($rad$)

$$rad_T = |rd_T|. \tag{13}$$

In the second analysis, we assess the gain of the mixed copula estimator over the mixed probability estimator. For this direct comparison of the two mixed distribution approaches we use their respective deviation

$$\Delta rd_T = \frac{Tmix - TmixC}{TmixC} \tag{14}$$

and its absolute value

$$\Delta rad_T = |\Delta rd_T|. \tag{15}$$

In the third analysis we want to gain deeper insight into the conditions under which the mixed copula approach improves the mixed probability estimator. For this purpose the influencing variables accounting for the difference of both estimators are

examined using multiple regression. The predictors considered include the catchment area $AREA$, the base flow index $BFI$, and the recession constant as defined in Tallaksen and Van Lanen (2004) and Gustard and Demuth (2008). Because runoff recession in the study area may be disturbed by freezing processes in winter, we chose to use the summer recession constant $REC7s$ based on recession periods of seven days or more, according to an exploratory assessment. Expecting the estimator to also depend on seasonality, we include the seasonality ratio $SR$ (quotient of mean annual summer and winter minimum flow)

and the variability measure of the circular seasonality index ($r$) in the analysis. All predictors have been screened for possible nonlinear correlations using the same procedure as in the companion paper, with a significant difference between linear and rank correlation used as a symptom of nonlinearity. From this preliminary assessment, we have found log-linear relationships of catchment area and the recession constant and decided to use the (decadic) log-transformed variables $log(AREA)$ and $log(REC7s)$ accordingly. For the seasonality ratio we use the absolute logs, termed $abs\_log(SR)$, as in the companion paper.

Finally, the seasonal correlation was entered as an absolute value ($abs\_Rho$), which appeared appropriate to deal with slightly negative correlations observed in a few catchments. The resulting predictors are listed in 2.

To test their significance, we use the common Student t-test for regression parameters at the $\alpha = 0.05$ level. We do this using a multiple regression model with main effects and two-way interaction terms. The parameters are checked for collinearity using variance inflation factors (VIF) and the adjusted coefficient of determination $R^2$. As this is a penalized measure of model

performance, a greater difference between adjusted and unadjusted $R^2$ will be interpreted as an indication of overfitting, and a low difference as indication of a well-determined model.



The data analysis was performed in R (R Core Team, 2021), using in particular the following packages: lfstat (Koffler et al., 2016), lmom (Hosking, 2022) and FAdist (Aucoin, 2022) for low flow statistics, drought2015 (internal package by Tobias Gauster), reshape2 (Wickham, 2007), ggplot2 (Wickham, 2016) for drought mapping, copula (Hofert et al., 2022) and 255 VineCopula (Nagler et al., 2022) for copula analysis, and EnvStat (Millard, 2013) and xtable (David B. Dahl et al., 2022) for distribution plots and tables.

**Table 2.** Overview of hydrologic predictors used in the regression model.

| Variable | Description | Unit |
|---|---|---|
| $abs\_Rho$ | Correlation of summer and winter events (log-transformed) | − |
| $r$ | Strength of seasonality | − |
| $log(AREA)$ | Catchment area (log-transformed) | $\mathrm{km}^2$ |
| $BFI$ | Base flow index | − |
| $abs\_log(SR)$ | Seasonality ratio (absolute logarithms) | − |
| $log(REC7s)$ | Summer recession constant (log-transformed) | $\mathrm{m}^3\mathrm{s}^{-1}\mathrm{d}^{-1}$ |

## 4 Results

### 4.1 Performance gain compared to the annual estimator

We first assess the performance gain of the mixed copula estimator compared to the annual estimator based on the non-260 cumulative and cumulative distributions of relative deviances across Europe (Fig. 3). Absolutely large values indicate a big gain of the mixed copula estimator over the conventional frequency analysis approach. Summary statistics of absolute deviations are presented in the upper panel of Table 3. From the graphs, the distribution is quite similar to that of the mixed probability estimator in Austria (Laaha, G., 2022, Fig. 3). For high return periods such as $T$=100 years, the spread is somewhat wider than that in the Austrian study, indicating a larger proportion of stations with high deviations. Apart from these extreme cases, 75 265 % of stations show an (absolute) performance gain of > 7 %, 41 % of stations > 50 % and 25 % of stations > 98.7 %. Minor differences can also be observed for lower return periods, such as the 20-year event, where again 25 % of stations show a performance gain of more than 33.1 %. In fact, we cannot assess at this stage whether the differences are due to the use of a different estimator or a different data set. But overall, the differences between the two studies appear to be small, suggesting similar performance of both mixed distribution approaches.

The companion paper has inferred from the Austrian study area that mixed distribution approaches should be relevant for a wide range of regimes and it is now interesting to assess this finding at the pan-European scale. Figure 4 shows such a mapping for the relative gains of the mixed copula estimator for moderate ($T$=2) and severe ($T$=100) low-flow conditions and demonstrates that this is indeed the case. Large values are scattered all over Europe, indicating mixture of summer and winter distributions to happen in all regions. For $T$=100 years the gains (up to 100 %) are most pronounced along the Alpine





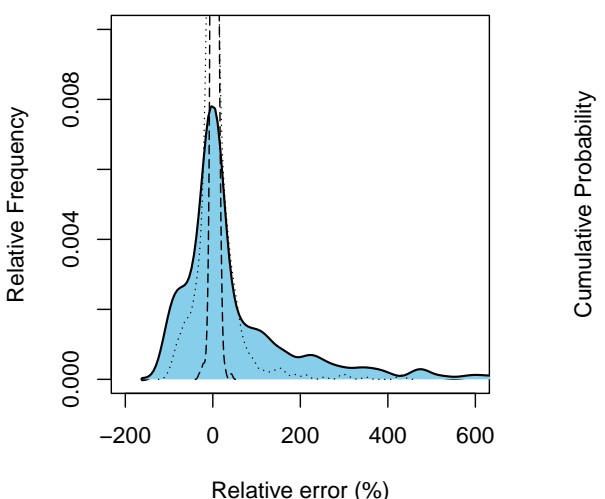
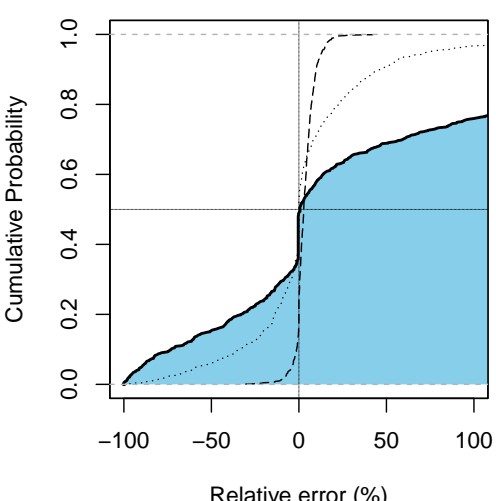

**Figure 3.** Uncertainty of the annual probability estimator as compared to the mixed copula estimator. Full line with blue shaded area refers to the 100-year event. For comparison the 20-years event (dotted line) and the 2-year event (dashed line) are shown

arch and in Fennoscandia, where the annual estimator tends to overestimate. Large gains can also be seen in the lowlands north of the Alps where the annual estimator tends to underestimate. For moderate low-flows such as the $T$=2 year event, a different pattern emerges. The gains are generally smaller (mostly up to 10 %, in rare cases up to 50 %) and reddish colours predominate, showing a general tendency of the annual estimator to overestimate. The gains of the mixed copula approach are most pronounced in the lowlands north of the Alps which is subject to cold climate. The gains are much smaller in the west,

which is subject to temperate climate. Altogether, the patterns suggest a large gain of the mixed copula estimator, making the method highly relevant for Europe as a whole.

Table 3 allows a more quantitative error assessment. For comparison, the relative absolute deviation of the mixed distribution estimator ($Tmix$) from Laaha, G. (2022) evaluated for the European data set is presented in the lower panel. The statistics confirm that the two mixed probability estimators, $TmixC$ of this paper and $Tmix$ of the companion paper, actually have

quite similar performances. For both estimators, the gain compared to the annual estimator decreases when the return period decreases. This is in line with Laaha, G. (2022), which found that the gain of the mixed probability estimator is substantial at high return periods but diminishes at low return periods. However, the difference between the two estimators increases at low return periods. While the performances are quite similar from 100 to 20 years, the median $rad$ for $TmixC$ and $Tmix$ are 4.78 and 5.45 for $T$=10 years and 3.48 and 7.84 for $T$=2 years, respectively. This may point to an interesting difference of the

approaches, which will be examined hereafter.





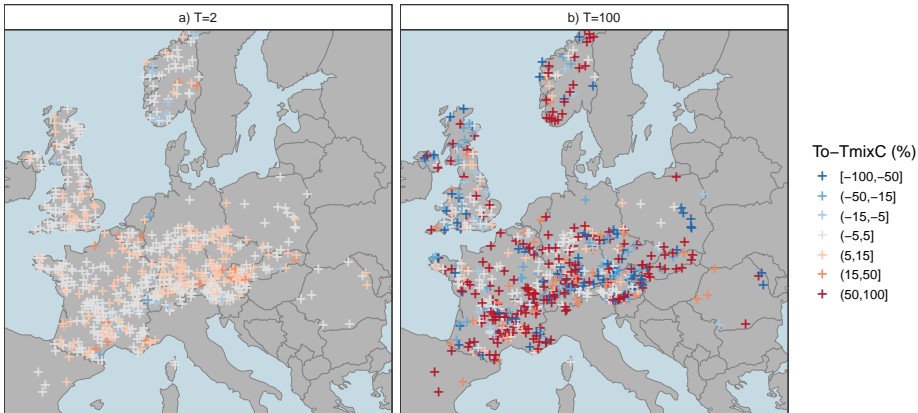

**Figure 4.** Relative deviation $rd$ of the mixed copula estimator from the annual probability estimator for a moderate low-flow event with $T$=2 year return period (left panel) and an extreme low-flow event with $T$=100 year return period (right panel). Blue colors indicate underestimation of the annual probability estimator, white color low difference between estimators, and red colors overestimation of the annual probability estimator.

**Table 3.** Relative absolute deviation ($rad$) of the mixed copula estimator ($TmixC$) from the annual probability estimate $T$ (in %). For comparison, the relative absolute deviation of the mixed distribution estimator ($Tmix$) from Laaha, G. (2022) evaluated for the European data set is presented below.

| Estimator | T (years) | Min. | 1st Qu. | Median | Mean | 3rd Qu. | Max. |
|-----------|-----------|------|---------|--------|------|---------|------|
| $TmixC$ | 100 | 0.00 | 7.00 | 41.43 | 123.05 | 98.69 | 2562.87 |
| $TmixC$ | 50 | 0.00 | 4.67 | 25.25 | 63.13 | 74.75 | 1231.49 |
| $TmixC$ | 20 | 0.00 | 1.96 | 11.21 | 25.41 | 33.07 | 432.93 |
| $TmixC$ | 10 | 0.00 | 0.94 | 4.78 | 11.95 | 15.84 | 167.62 |
| $TmixC$ | 2 | 0.00 | 1.10 | 3.48 | 4.67 | 6.67 | 42.15 |
| $Tmix$ | 100 | 0.00 | 7.06 | 41.31 | 124.25 | 98.69 | 2562.87 |
| $Tmix$ | 50 | 0.00 | 4.59 | 25.55 | 64.04 | 75.59 | 1231.49 |
| $Tmix$ | 20 | 0.00 | 1.96 | 11.68 | 26.26 | 34.62 | 432.93 |
| $Tmix$ | 10 | 0.00 | 0.87 | 5.45 | 13.17 | 18.90 | 167.62 |
| $Tmix$ | 2 | 0.00 | 2.31 | 7.84 | 9.90 | 15.91 | 57.44 |

## 4.2 Performance gain compared to the mixed estimator

In the second assessment we evaluate the relative difference of the two mixed distribution approaches in greater detail. Figure 5 and Table 4 show the relative performance gain of the mixed copula approach over the mixed probability estimator, and Table





**Table 4.** EU: Relative deviation ($\Delta rd$) between the mixed and mixed copula probability estimator (%)

|     | Min.   | 1st Qu. | Median | Mean  | 3rd Qu. | Max. |
|-----|--------|---------|--------|-------|---------|------|
| 100 | -8.25  | 0.00    | 0.00   | -0.29 | 0.00    | 0.00 |
| 20  | -13.88 | -0.65   | 0.00   | -0.92 | 0.00    | 0.00 |
| 2   | -22.54 | -7.64   | -2.93  | -4.75 | -0.44   | 0.00 |

5 gives the absolute numbers. As indicated above, the added value of using the copula-based estimator is only minor for the
100 year event. However, the gain for the moderate low-flow event is considerable. Relatively low effects are still observed
for western Europe with typical deviations of less than +/-5 % in the majority of catchments and between -5 and -15 % in
the remaining catchments. Such low effects are also observed in the high Alps, where the low-flow seasonality is strong and
the gain of the mixed estimator was found to be low (Laaha, G., 2022). Higher differences occur in the northern and southern
forelands of the Alps, as well as in central and eastern Europe. In these regions the relative deviation of both estimators are
about of -15 to -23 %.

Interestingly, the spatial patterns of Fig. 5a appear quite similar to those of the Fig. 4a, suggesting that the difference of the
two estimators is generally higher where the gain of the mixed distribution approaches are large. This "size effect" may point
to some uncertainty of the mixed probability estimator, which can be corrected using the mixed copula approach. In fact, the
difference is always negative or zero (Table 4). Assuming $TmixC$ as the true model, $Tmix$ can be regarded to underestimate
the return period, and $TmixC$ to correct this underestimation.

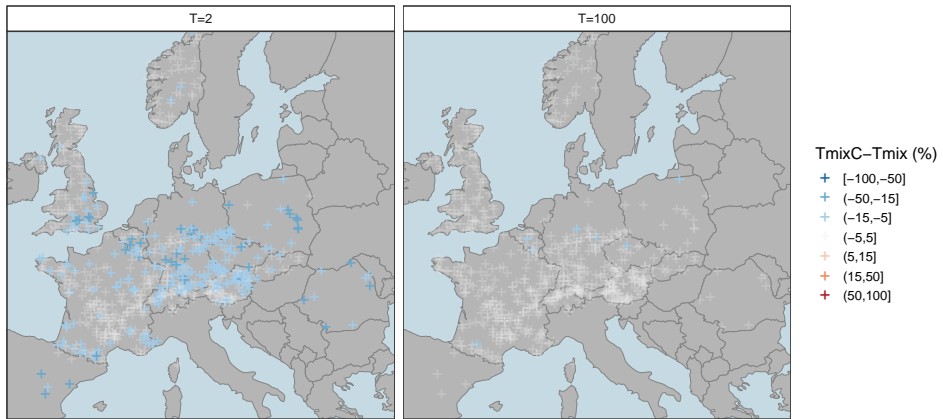

**Figure 5.** Relative deviation $\Delta rd$ of the mixed copula estimator from the mixed probability estimator for a moderate low-flow event with
$T$=2 year return period (left panel) and an extreme low-flow event with $T$=100 year return period (right panel). Blue colors indicate under-
estimation of the mixed probability estimator and white color low difference between both estimators.





**Table 5.** EU: Relative absolute deviation ($\Delta rad$) between the mixed and mixed copula probability estimator (%)

|     | Min. | 1st Qu. | Median | Mean | 3rd Qu. | Max. |
|-----|------|---------|--------|------|---------|------|
| 100 | 0.00 | 0.00 | 0.00 | 0.29 | 0.00 | 8.25 |
| 20  | 0.00 | 0.00 | 0.00 | 0.92 | 0.65 | 13.88 |
| 2   | 0.00 | 0.44 | 2.93 | 4.75 | 7.64 | 22.54 |

### 4.3 Conditions that influence the additional performance gain

In a deeper analysis we assess the variables controlling this additional performance gain at low return periods ($T = 2$) by multiple regression. The previous analysis have pointed to a "size effect", which is known to predominate in statistical regressions. In order to emphasise on the additional effects a two-step approach is adopted. We first fit a simple linear regression model to
adjust for the size effect of the mixed probability estimators, using the relative absolute performance gain of the mixed copula estimator $TmixC$ as a predictor. We then assess the residuals of this model by multiple linear regression with main effects and two-fold interaction terms.

We first analyse the predictors as main effects. This is a simplified assessment as interdependence of the predictors is not covered and latent interactions may be misclassified as main effects. Nevertheless, the model is useful to give a first indication
of variables that have some predictive value. The resulting model was carefully checked for over-fitting and found to be well determined. The summary statistics presented in Table 6 show that the four predictors representing seasonal correlation $Rho$, seasonality $r$, catchment $AREA$, and base flow $BFI$ are highly significant. The baseflow recession constant is significant as well, but no significant effect of the seasonality ratio could be found. However, these predictors are intercorrelated with the previous predictors and discarding them leads to no decrease of the model fit (with $R^2 = 0.58$ for all models). They are
therefore of little significance for the model and dropped from further analysis.

The same procedure was subsequently used to fit a model with main effects and two-fold interaction terms. Care had to be taken for possible overfitting as the full model had high VIF values (up to 59) for all predictors. The model was therefore subjected to a stepwise variable elimination, which reduced the model by most of the insignificant predictors, thereby resulting in VIFs that were considered acceptable. As the fit improved and overall significances did not change we found the full model
well justified. The model has an $R^2$ of 0.68 and an adjusted $R^2$ of the same value, which suggests that it is not overfitted and is well suited to capture the main differences between the two probability estimators. The summary statistics of the model are shown in Table 7. Interestingly, the model only uses the seasonal low-flow correlation ($Rho$) as a main effect, which is well in line with the basic intention of the approach to improve the estimation in case of correlation. The catchment $AREA$, strength of seasonality $r$ and catchment storage (characterised by the $BFI$) exhibit significant interactions with the seasonal correlation.
The bottom lines of the table show two model alternatives where $r$ and $BFI$ were replaced by their correlated counterparts $abs\_log(SR)$ and $log(REC7s)$ respectively. Both variables are also significant predictors of seasonality and storage, although in the case of $abs\_log(SR)$ with reduced model performance. The analysis suggests that the performance gain of the mixed





**Table 6.** Multiple regression summary statistics (main effect model) of the relative absolute deviation ($\Delta rad$) between the mixed and mixed copula estimator for the $T$=2 years return period.

|  | Estimate | Std. Error | t value | Pr($>$|t|) |
|---|---|---|---|---|
| (Intercept) | 2.4974 | 0.8225 | 3.04 | 0.0025 |
| abs_Rho | -13.7881 | 0.6210 | -22.20 | 0.0000 |
| r | 9.8974 | 0.8306 | 11.92 | 0.0000 |
| log(AREA) | -0.5221 | 0.1661 | -3.14 | 0.0017 |
| BFI | -2.9458 | 1.1006 | -2.68 | 0.0076 |
| abs_log(SR) | -1.3395 | 0.6455 | -2.07 | 0.0383 |
| log(REC7s) | 0.1209 | 0.4766 | 0.25 | 0.7999 |

**Table 7.** Multiple regression summary statistics (main effects and two-way interaction terms model) of relative absolute deviation ($\Delta rad$) between the mixed and mixed copula estimators for a return period of $T$=2 years. Insignificant effects were removed by automatic backward stepwise variable selection. The bottom lines show two model alternatives where either $r$ or $BFI$ were replaced by their correlated counterparts, $abs\_log(SR)$ (resulting in an $R^2$ of 0.55) and $log(REC7s)$ ($R^2$ of 0.67), respectively.

|  | Estimate | Std. Error | t value | Pr($>$ |t|) |
|---|---|---|---|---|
| (Intercept) | 5.2518 | 0.2226 | 23.59 | 0.0000 |
| abs_Rho | -28.3210 | 1.9945 | -14.20 | 0.0000 |
| r : abs_Rho | 33.8512 | 1.6223 | 20.87 | 0.0000 |
| BFI : abs_Rho | -12.2736 | 1.7464 | -7.03 | 0.0000 |
| log(AREA) : abs_Rho | -1.0727 | 0.3081 | -3.48 | 0.0005 |
| log(REC7s) : abs_Rho | -5.6570 | 0.9174 | -6.17 | 0.0000 |
| abs_log(SR) : abs_Rho | 9.6087 | 0.9476 | 10.14 | 0.0000 |

copula over the mixed distribution estimator can be well explained by the seasonal correlation and catchment characteristics. The model is thus a suitable tool to analyse these effects in more detail.

Figure 6 depicts the effects of the considered predictors as partial regression lines together with standard errors and partial residuals. The plots allow a direct comparison of effect size between predictors with various range of variability. The results show how much the weak performance of the mixed probability estimator at low return periods is due to seasonal correlation. Catchments with strong seasonal correlation generally show a stronger underestimation of the mixed probability estimator than weakly correlated catchments (Fig. 6a). The copula estimator thus effectively reduces the errors of the mixed probability

estimator caused by correlation of summer and winter events as one would actually expect. However, the correlation effect is





modulated by interactions with seasonality, catchment size and storage (Fig. 6b-d). On the one hand, the underestimation is less pronounced for catchments with strong seasonality, where the gain of the mixed estimator was found to be low (Laaha, G., 2022). This is indicated by a clear positive partial effect of $r$ (Fig. 6b). On the other hand, the correlation effect is enhanced for catchments with a high discharge share from stored sources and a large catchment size, as shown by the negative partial
effect of the $BFI$ and $AREA$ (Fig. 6c-d). These results complement the findings from the exemplary catchments of Fig. 1 and generalise them to a wide range of European regimes.

It is finally interesting to assess the spatial patterns of the effects at the pan-European scale (Fig. 7). The upper panels illustrate nicely the antagonistic nature of seasonal correlation and strength of low-flow seasonality as outlined above. Catchments north of the Alps are prone to strong seasonal correlation which contributes to an underperformance of $Tmix$. This affects the
mixed low-flow distribution, unless there is strong seasonality as well, so that only one distribution dominates. Such strong (summer) seasonality is typical for temperate climate in western Europe, where the gain of the mixed copula estimator is consequently small (Fig. 7a). The combination of strong correlation and strong summer seasonality corresponds to Type (c) in Figure 1. However, in regions with weaker seasonality, a stronger mixture of distributions can be observed (Laaha, G.. Under such conditions, a strong seasonal correlation has has a strong effect on the mixed distribution estimator, and the gain of the
mixed copula estimator is correspondingly high. Such a constellation typically occurs in mid-mountain regions north of the Alps (e.g. Bavaria and Ore Mountains (GER), Tatra mountains (SLO), Carparthians (PL), and the foothills of the Pyrenees in Southern France). The combination of strong correlation and weak to moderate summer seasonality corresponds to Type (b) of Figure 1, for which a Bavarian gauge is the showcase example. Finally, the lower panels of Figure 7 depict the interactions of base flow index and catchment area, which show much lower effects. Clearly, these variables have a lower contribution to the
underperformance of the $Tmix$. The $BFI$ shows some similarity with seasonal correlation and therefore enhances the correlation effect on the estimator. The $AREA$, however, shows very small values and should have little impact on the performance of the mixed probability estimator.





**Figure 6.** Termplot of multiple regression (main effects and two-way interaction terms model) of relative absolute deviation ($\Delta rad$) between the mixed and mixed copula estimators for a return period of $T$=2 years.



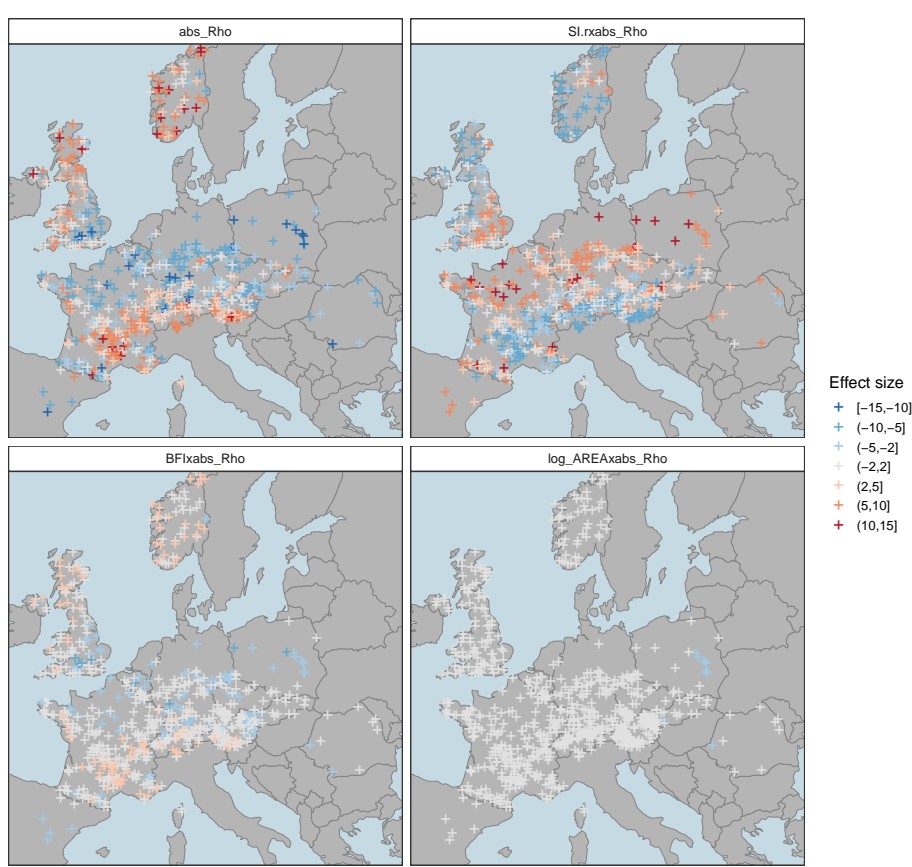

**Figure 7.** Effect size of multiple regression (main effects and two-way interaction terms model) of relative absolute deviation ($\Delta rad$) between the mixed and mixed copula estimators for a return period of $T$=2 years.





## 5   Discussion and conclusions

This second paper of a two-part series presents the mixed copula estimator for low-flow frequency analysis. The method
provides an extension of the mixed distribution approach of the companion paper (Laaha, G., 2022), to deal with dependency
of seasonal events. The paper formulates the copula-based estimator for seasonal minima series and examines its statistical
properties in a hydrological context. We carried out a detailed assessment of the method to show its performance compared to
the mixed distribution approach.

As a starting point, the distributional characteristics of the mixed copula estimator were assessed by frequency plots of
archetypal examples representing contrasting low-flow regimes. Differences to the mixed estimator were always observed in
the upper part of the distribution, but this is less relevant for low-flow frequency analysis. At severe low-flow events, the
differences are very small. For mild to moderate events that fall between these cases, the differences between the two mixed
distribution approaches are determined by the interaction of seasonality and seasonal correlation.

In the subsequent step, the performance gain was evaluated based on a larger pan-European data set. We first assessed the
performance gain compared to the annual estimator. Both mixed distribution approaches show a similar performance at high
return periods. This emerges from a comparison of the two estimators for the $T$=100 year event. However, the difference
between the two estimators increase at low return periods. This can be seen in the $T$=2 year event, where the gains of the mixed
copula approach are most pronounced in the lowlands north of the Alps which is subject to cold climate. The gains are much
smaller in the west, which is subject to temperate climate. Altogether, the patterns suggest a large gain of the mixed copula
estimator over the annual probability estimator, making the method highly relevant for Europe as a whole.

We then assessed the performance gain of the mixed copula approach over the mixed distribution approach in greater detail.
The analysis show that the gain for the $T$=100 year event is indeed minimal. However, the gain for mild events is considerable
in some of the catchments, with relative deviation of -15 to -23 % in the most affected regions. The differences increase with
the performance gain of the mixed over the annual estimator. This points to a "size effect" that can be interpreted as some
inherent uncertainty of the mixed probability estimator, which is corrected by accounting for seasonal correlation. In fact, the
differences are always negative or zero, so that $Tmix$ shows the tendency to underestimate the severity of the event.

The variables controlling this additional performance gain at low return periods were further assessed by multiple regression.
The analysis suggests that the performance gain of the mixed copula over the mixed distribution estimator can be well explained
by the seasonal correlation and catchment related characteristics. We can interpret the model such that the copula estimator
is indeed effective in reducing the errors of the mixed probability estimator caused by the correlation of summer and winter
events. However, the correlation effect is modulated by interactions with seasonality, catchment area and storage. Effect maps
show the antagonistic nature of seasonality and seasonal correlation at the European scale, with weak seasonality leading to a
high potential for correction and strong seasonal correlation reinforcing the need for its consideration. These results are well
in line with findings from the exemplary catchments and generalise them to a wide range of European regimes. Storage and
catchment size only have a minor effect on the performance gain of the mixed copula estimator and appear to play a subordinate
role.



Although the differences found are high on relative terms, they mostly occur at low return periods. This raises the question of whether it is practical to use the more complex copula-based estimator, or whether the simpler mixed distribution approach will suffice. Some indications were already given in the results section, where regions with large differences and hence a stronger correction were identified. These relate to mid-mountain regions in cold and temperate climate where rivers have strongly mixed low-flow regimes. In these regions, the use of the mixed copula estimator always appears to be beneficial.

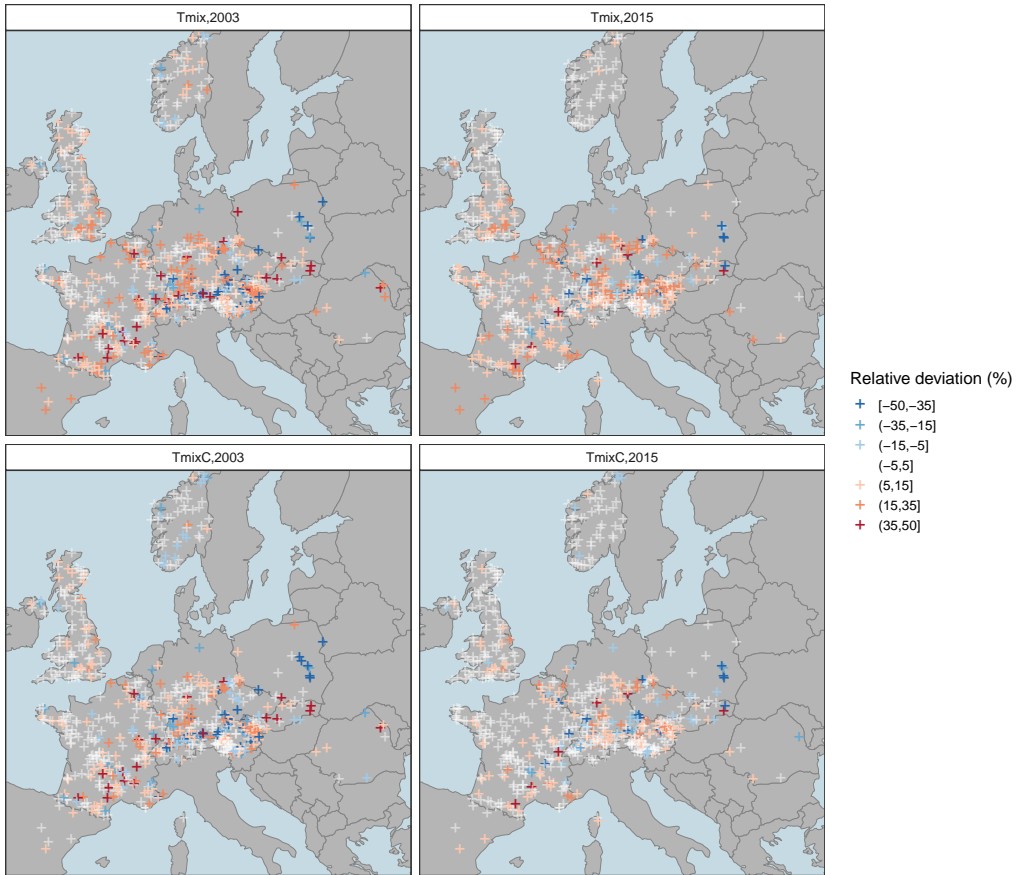

**Figure 8.** Implications of estimation uncertainty for drought event mapping, illustrated for the European cases 2003 (left) and 2015 (right). Shown are the relative deviations $rd$ of the mixed probability estimator (upper panels) and the mixed copula estimator (lower panels) from the conventional annual estimate.

Another aspect of practical relevance is the impact of uncertainties on a Europe-wide drought mapping, which is essential for the assessment of major events. Such mapping is often based on return periods that are traditionally obtained by the annual minima approach. This was also the case in our own study of the European 2015 event (Laaha et al., 2017). The magnitude of inherent estimation uncertainties with respect to the two enhanced estimators can be seen in Fig. 8. The panels depict the relative deviations of the mixed probability estimator (upper panels) and the mixed copula estimator (lower panels), as compared to the

annual estimates for the European 2003 and 2015 cases. Two interesting aspects can be observed. First, the effect of using a mixed probability estimator is again greater the more severe the spatial event. This is suggested by the comparison of the 2003 and 2015 cases, with the event of 2003 having a greater spatial extent and severity. Second, the relative deviations of $Tmix$ are

much larger than that of the mixed copula estimator $TmixC$. This is due to our finding that the $Tmix$ tends to overcorrect the uncertainties of the annual estimator in a number of catchments, leading to an underestimation of drought events. The mixed copula estimator reduces this underestimation in most of these cases, and the correction is particular visible in regions where a strong seasonal correlation with weak seasonality is superimposed. This suggests that the mixed distribution approaches can indeed be relevant for drought mapping, with the mixed copula approach providing the most accurate estimate.

Despite all the favorable features of the mixed copula estimator, it should be emphasized again that the difference between the two mixed distribution approaches is not large and mainly concerns the low return periods. It is therefore unlikely that severe events in their entirety, i.e., in terms of their spatial extent and severity, will be misclassified by the mixture distribution estimator. Differences are more likely at the regional level where drought conditions may be overlooked. Such regional view, however, is vital for drought monitoring, where the greater accuracy of the mixed copula estimator will allow for a more

accurate assessment of drought severity.

Overall, we conclude that the two mixed probability estimators are quite similar, and both are more accurate as the annual minima approach. In regions with strong seasonal correlation the mixed copula estimator is most accurate and should be preferred over the mixed distribution approach.

*Code availability.* The code will be made available in an updated version of R software package 'lfstat' via the CRAN repository.

*Author contributions.* The author is responsible for all aspects of the analysis and the manuscript.

*Competing interests.* No competing interests.

*Acknowledgements.* Data provision by the Hydrographical Service of Austria (HZB) was highly appreciated. I am grateful to Tobias Gauster for providing a dedicated R environment for European drought mapping (through the drought2105 package). This research is a contribution to the UNESCO-IHP VIII FRIEND-Water program.




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
