# Peer review of "A mixed distribution approach for low-flow frequency analysis – Part 2: Comparative assessment of mixed probability vs. copula-based dependence framework"

_Hydrology and Earth System Sciences, 2022_

## Referee Comment (RC2)

The paper proposes a mixture distribution approach to analyze the frequency of low flows considering the seasonality of river discharges and implements the same to investigate low flow frequency at representative German and Austrian catchments. This is a part of a companion work, where part I show the seasonality and performance, wherein part 2, the comparative assessment, is shown using a copula-based dependency framework. The comments on the manuscript are summarized below:

1. Throughout the manuscript (See line 70 for example), the authors have claimed he is not aware of any study that used copulas to consider seasonal dependence in a mixed distribution approach. Ganguli and Reddy (2014) developed copula-based ensemble drought prediction models with up to 3 months' lead time considering the seasonality of SPI at a 6-month accumulation window. Two variants of drought forecast models were proposed: a single ensemble drought prediction model without seasonal partition and separate models for each of the four seasons in a year, combining them to constitute a yearly forecast model. The analyses showed that the seasonal prediction model performs better as compared to the model without seasonal partition. In addition, the incorporation of a copula-based conditional framework helps to provide an estimation of uncertainty.

2. Eqs. 2, 6-8 representations of random variables are not correct. In fact, for summer, the random variables are drawn from the population, $S = \{s_1, s_2, \ldots, s_n\}$. Likewise, for winter, the random variables are drawn from the population, $W = \{w_1, w_2, \ldots, w_n\}$. The probability of a low-flow event with magnitude during summer, $s$ and winter, $w$ seasons to be represented as $F_S(s)$ and $F_W(w)$, respectively. Further, typically subscript/sample space is shown using capital letters, whereas the argument/individual random variables are shown using small letters with the same notations. They cannot be different, for example, $q$ as mentioned in the manuscript even after considering seasonal stratifications.

3. Line # 105, How is the copula parameter $\theta$ is, estimated? There is no direct relationship between Spearman's $\rho$ and $\theta$ for the Gumbel-Hougaard copula. However, such a relationship exists for Kendall's $\tau$ and $\theta$. Also, it would be good to see the gauge-wise performance of Gumbel-Hougaard copula for the summer and winter seasons either in the SI section or in the main manuscript to see how credible are the copula performance in modelling seasonal dependence.

4. Eq. 9, same error for random variable representation, $P_Q(q)$ instead of $P_m(q)$.

5. At first, Eq. 11 should appear, followed by Eq. 10. Again, representations of random variables should be corrected considering seasonal partition in line with Eq. 9 and others.

6. Line #155, there is a subtle difference between the two estimators.

7. Line # 168, "We note that this behavior…

8. Line # 173, how seasonality and seasonality ratio are determined in this study? Also, please discuss the associated implications of each of these indices in brief.

9. Line # 190 and paragraph therein: However, the copula-based approach is expected to preserve seasonal dependence patterns apart from the fact that they consider the marginal distribution of any form. On the other hand, the mixed distribution assumes only one type of probability distribution.

10. Line 195, could you provide a list of gauges, WMO ID, their latitude/longitude, catchment area, and years of available records in SI as part of the reproducibility of the work?

11. How the low flow is estimated in this study? Whether the constant/variable threshold approach is implemented to detect low flow signal.

12. Line 215: How relevant would be seasonal exceedance probability estimation since the concept of return period revolves around the sampling of annual and partial duration series?

13. Eq. 13, the relative absolute deviation will not show any over/underestimation effect. Therefore, the absolute unit would be more beneficial.

14. Fig. 3 caption; does the uncertainty of the annual probability estimator considers all catchments across the pan-EU scale?

15. Line # 288: the large differences between two estimators at low return period is consistent with the differences in quantile estimates between the annual maxima/minima and partial duration approach. In fact, the difference in return period estimation in Annual maxima/minima vs. partial duration series is generalized using a simple exponential relationship. Can the author derive such kind of generalized formula for the given EU catchments?

16. For Figs. 2, 4-5, and 7-8, please use a continuous color bar at the bottom of the figures and show the color discretization.

17. Table 3. The minimum quantile is always zero, indicating water level is always less than the stage recorder during the low flow period; therefore, instead of furnishing information on the Min quantile, the $1^{st}$ ($25^{th}$ percentile), median, and $3^{rd}$ quantiles, including the interquartile range, would suffice. In fact, the IQR would show the catchment-wise variability in low flows for each estimator.

18. Line # 322, VIF is not defined earlier.

19. Line # 344: This suggests sensitivity towards BFI, which is, in turn, the function of catchment soil types and the availability of water bodies nearby.

20. Line 353: Laaha, 2022?

21. Line 386: the uncertainty could also stem from estimated copula parameters and the uncertainty due to marginal distribution. Therefore, sample lengths have a profound impact

on multivariate distribution. A list of available sample lengths is to be presented. For credible assessment of multivariate risk, sample lengths need to be at least more than 25 or larger.

22. Line 388: authors have pointed influence of catchment area, BFI, and to some extent, climate; however, terrain attributes, soil types, and land use/land cover do also have a profound impact on drought seasonality, persistence and recovery pattern.

23. Line 400: The pronounced differences in quantiles, mainly at low return periods, are consistent with annual vs. partial duration series, wherein the differences tend to diminish for high return period events. A generalization of this would add value.

24. This study is useful at a regional level focusing only EU zone. A discussion on how this study would add value for other parts of the globe, for example, in monsoon-dominated regions (South Asia and Africa) where marked seasonality is pronounced or in areas with relatively stable climate with subtle climatic variability (for example, sub-tropics) would be beneficial for audiences across the globe.

25. Last but not least, the study assesses the performance of mixed distribution vs. copula-based dependency framework for modelling low flows accounting seasonality. This message is not well reflected in the title. The title could be tuned in that direction, for example: A bivariate approach for low-flow frequency analysis considering seasonality Part 2: Comparative assessment of Mixed Probability vs Copula-based Dependence Framework.

**26. A Minor comment:**

Line 41: Ganguli and Reddy (2012) presented a bivariate risk assessment framework for meteorological droughts in Saurashtra and Kutch regions of Gujarat state in India. Based on the tail dependence measure, the Gumbel-Hougaard copula emerged as the best model for modeling joint dependency between drought severity and duration. The comparative assessment of traditional bivariate distributions, such as bivariate log-normal and bivariate logistic models relative to copulas suggested that the extreme value family of the Gumbel-Hougaard copula was better suited for the area.

**References:**

Ganguli, P. and Reddy, M. J.: Risk Assessment of Droughts in Gujarat Using Bivariate Copulas, Water Resour Manage, 26, 3301–3327, https://doi.org/10.1007/s11269-012-0073-6, 2012.

Ganguli, P. and Reddy, M. J.: Ensemble prediction of regional droughts using climate inputs and the SVM–copula approach, Hydrological Processes, 28, 4989–5009, https://doi.org/10.1002/hyp.9966, 2014.

---

## Author Response (AR1)

*Author's response*

*I would like to thank the editor and the reviewers for their valuable time spent for reviewing my MS. Please find my responses below.*

*Response Reviewer #1*

*The work „A mixed distribution approach for low-flow frequency analysis – Part 2: Modeling dependency using a copula-based estimator" by Gregor Laaha proposes a seasonal mixture distribution for low flows by explicitly taking account the dependency between the seasons using a copula approach. The applicability and main differences to common approaches are demonstrated for a large European data set and attributed by regression models using catchments and seasonality indices as explaining variables.*

*The manuscript is well written and logically structured. The research topic is highly relevant, as it extends the existing seasonal mixing approaches for floods to the low-flow case and takes into account the important aspect of dependence between winter and summer low flows. Especially this extension makes it relevant to hydrological sciences, as it is a novel approach considering the special characteristics of low flows.*

*I have a few comments that might help the readership to better follow the stream of thoughts and provide some background information. With these minor changes, the manuscript should be accepted for HESS.*

**RE: I would like the reviewer for his positive feedback and his valuable commends on the MS. Please find my point-by-point responses below. The line numbering in my responses refers to the revised version of the MS, unless otherwise indicated. The reviewer comments are presented in italic format, my responses in roman typeset. Additionally, to facilitate navigation through our responses I report in blue direct quotes from the MS (i.e. unchanged text) in blue color Changes (new text lines) in the revised MS are highlighted in red color.**

*1. I have concerns when using the term "accurate" when comparing different statistical models (e.g. in the abstract and the last lines of the conclusion). I can understand the idea of comparing the models by estimating the differences in the low flow quantiles. As always for statistical models in hydrological statistics, it is almost impossible to compare the results to the "true" value, as we simply cannot observe this. We are limited to the observations we have and can only compare the results of the models. Hence, the term "more accurate" for one model compared to another is not suitable, even if one says that statistically the new model is more reasonable (to which I absolutely agree). The only way to justify something like accuracy would be simulations, as in this case the truth is known. I therefore recommend to weaken the statements in the manuscript and omit terms like "more accurate". Instead, the use of the term "gain" seems to be a very good idea to me and should be done throughout the whole manuscript. Similarly, the terms "error" (Figure 3) as well as under- and overestimation (e.g. ll. 275-276) seem to be misplaced here. I would also highly appreciate a simulation study that clearly demonstrates the benefits of the copula mixture model, but I can understand that this would increase the length of the manuscript too much.*

RE: Thanks for this comment. I completely agree with this statement and it was actually my intention to avoid too strong statements, as the true distribution is indeed unknown, and superiority of an

estimator over another is derived from theoretical considerations. Following the advices, I have revised the wording accordingly.

For example, I have rephrased the text sections and passages pointed out by the reviewer as follows:

Abstract:

L. 11: The analysis shows that the difference for the 100-year event are actually minimal. However, the differences for 2-year events are considerable in some of the catchments, with a relative deviation of -15 to -23 \% in the most affected regions.

L. 18: We conclude that the two mixed probability estimators are quite similar, and both are conceptually more adequate than the annual minima approach for mixed summer and winter low-flow regimes. In regions with strong seasonal correlation the mixed copula estimator appears most appropriate and should be preferred over the mixed distribution approach.

L. 275-276 (old numeration): For $T$=100 years the gains (up to 100 \%) are most pronounced along the Alpine arch and in Fennoscandia, where the annual estimator yields higher return periods than the mixed copula approach. Large differences can also be seen in the lowlands north of the Alps where the annual estimator tends to produce lower estimates.

Last lines of conclusions:

Such regional view, however, is vital for drought monitoring, where the statistically more reasonable concept of the mixed copula estimator should allow for a more accurate assessment of drought severity. Overall, we conclude that the two mixed probability estimators are quite similar, and both are conceptually more adequate than the annual minima approach. In regions with strong seasonal correlation the mixed copula estimator appears most appropriate and should be preferred over the mixed distribution approach.

*2. The author states that the Gumbel-Hougaard copula is used "as we are modeling extreme values". There exist many copula models for this purpose and I cannot see why only this should be suitable, especially for an application to whole Europe. I suggest to use a goodness-of-fit criterion such as AIC/BIC to select the best-fitting copula from a sample of possible models. It is not a priori clear to me why there should always be the same dependence structure for all catchments.*

RE: The choice of the Gumbel-Hougaard model is based on a literature review from which I inferred its general suitability to model dependencies between extreme value distributions. Studies have either shown the direct suitability of the model (e.g. Klein et al., 2011; Poullin et al., 2007) or that the differences between different extreme value copula models to be mostly negligible (Genest and Favre, 2007). This was also confirmed for the data at hand in a preliminary analysis, and as the purpose of the study was less a comparative assessment of copula models than demonstrating the general suitability of mixed distribution approaches with respect to seasonal correlations, the priority was given to use the same model for all stations rather than using different models across the study area.

In response of the comment, I have checked the validity of the assumed Gumbel-Hougaard copula in greater detail, and assessed the sensitivity of the results of the study when using a possibly more adequate model. A first GOF assessment (Cramer-von-Mises test implemented in gofCopula of the copula package) shows that the GH copula was rejected in 15% of stations which is somewhat higher than the theoretical rejection rate of the hypothesis test (alpha = 0.05). In a further assessment a cross-validated log-likelihood criterion showed that the GH copula performs best among five EV

copulas (according to the number of best performing cases), whereas the Gumbel copula has only above-average rank (5.1) among eight other non-extreme value copula families and is outperformed by the Plackett (average rank 6.5) and Frank copula (average rank 6.5). This suggests a mixture of the three very contrasting models to be more adequate, given that there is no clear theoretical preference between them.

Following the suggestion of the reviewer I revised the paper using the station-wise best performing of three models (Gumbel, Plackett and Frank copula). In addition, I used the Independence copula to model 30 cases with a negative Rho which are all insignificant and not plausible from a drought generation process perspective. The method of copula family selection is described in the new Section 3.2, and changes were made to the theoretical concept of the mixed copula estimator so that it is not limited to a GH copula. (L. 116).

3.2 Choice of the copula family

As the copula family is a-priory unknown we screened a number of alternative families and compared them using a copula information criterion. Here we used function xvCopula of the R package copula to compute the crossvalidated log-likelihood of \citet{gronneberg_copula_2014} defined in equation (42) therein. When computed for several parametric copula families, it is meaningful to select the family maximizing the criterion. The analyses revealed for the pan-European data set that the Gumbel-Hougaard family performs best among five extreme value copulas, but ranks only slightly above average among eight non-extreme value copula families. In particular, the Gumbel-Hougaard copula (average rank of 5.1) was outperformed by the Plackett copula (average rank 6.5) and the Frank copula (average rank 6.4). As the three models have a contrasting structure and the true model structure is a-priory unknown, it was decided to use the station-wise best fitting of the three models for the pan-European assessment. In addition, we used the Independence copula to model 30 cases with a negative $Rho$, which was insignificant in all cases. Among the considered models, the Plackett and Frank families are each superior in 308 and 241 cases, while the Gumbel-Hougaard family fits best in 200 cases. Once for a station the family was selected, its parameter $\theta$ was fitted using the maximum-likelihood estimator. The resulting copula was finally inserted in the mixed copula estimator (Eq. \ref{equation_6}).

L 116: However, as the adequate model is a-priory unknown alternative copula families can be considered as well. In this case the best suited model can be chosen based on a copula information criterion \citep{gronneberg_copula_2014}.

In the remainder of the paper, all results have been adjusted accordingly. I remark that the results are not very sensitive to these changes so that the conclusions did not change. The regression effects of the considered predictors are even somewhat stronger and lead to clearer patterns across the study area, which supports the use of the best of different copulas at each case.

*3. After the model description, the manuscript goes right into the details of the results. I am missing at least a short description of the data. As one major aspect here is the dependence between winter and summer low flows, it is of high interest how these events have been defined. What is done in a case when a summer low flow reaches into winter or there is no clear distinction between summer and winter? How is winter and summer defined and are the same seasonal thresholds used for whole Europe? These are aspects which make the application of seasonal mixture models to low flows so much harder compared to floods. Moreover, they motivate this whole study.*

RE: A short description of the data has been added and also a statement about the range of the seasonal Rho values showing the various degrees of dependence between summer and winter lows (first and second paragraph of Section 3.1).

L. 216: These include 673 series with complete records and further 80 series with a record of at least 30 years and 29 records of at least 25 years. In addition, the data of the year 2015 are used to illustrate the sensitivity of estimation methods for major drought events. The data collection covers catchments from 86 to 104571 $km^2$ with streamgauge elevations from 2 to 1995 $m.a.s.l.$ and a mean annual 7-day low-flow discharges between 0.0002 to 1108.5 $m^3s^{-1}$.

L. 225: The values of the seasonal Rho ranges between 0 and 0.90 reflecting a great variability between completely independent and strongly dependent seasonal series. There are also 30 cases with negative, though not significant correlations which are therefore considered seasonally independent.

Concerning the questions about how winter and summer low flows are defined and how to deal with the case that low flows reach from one season into the other, and what seasonal thresholds are used: I think there might be a misunderstanding of the event definition used in this paper (and I realize that I was not clear enough right from the beginning of the paper defining the low flow event, which I will improve in the introduction of the revised MS): I am not analyzing low-flows based on thresholds (i.e., Yevjevich's threshold level approach), where the effects of droughts continuing from one season to the other make a clear attribution of events hard to obtain. Such analysis would indeed be very sensitive to the defined summer and winter seasons and seasonal thresholds used across Europe.

However, I am focusing here on low-flow magnitude as described by annual and seasonal mina series, where the seasonal attribution of events into a winter and summer type are straightforward. Earlier studies (such as Laaha et al., 2017) have shown that defining a winter season from December to April is appropriate to attribute frost-generated annual minima uniquely to the winter season, and remaining events (generated by meteorological water-balance deficits) to the summer season. Ongoing events at the end of a season are explicitly considered in this paper by their seasonal correlation and enter into the copula model according to their station-specific relevance. For these reasons, I would argue that mixed distribution approaches are more straightforward for low-flows than for flood peaks where attribution to process types is much harder to obtain than for low-flows, which is also reflected in the much clearer mixed distribution plots for low-flows than those published for floods.

To improve the clarity of the paper, the definition of the low-flow index has been clarified right from the beginning of the paper (see L. 26, 29, 34, 138 of the revised MS).

*4.    The indices under consideration should be explained. For example, it is not clear what SR or the circular seasonality index r are. In Table 1 it is stated that SR is the "ratio between mean summer and mean winter low-flow." If this is correct, why is it an indicator of seasonality? If there are few extreme low flows in summer and otherwise many large low flows while in winter there are many small low flows, the indicator would be highly impacted by the extreme low flows in summer. Wouldn't it be more reasonable to use the proportion of summer (or winter) low flows in the annual maximum series to investigate which has a greater impact on the AM? The index r only appears in Table 1 but is not mentioned in the text.*

RE: In response to the comment, the following text was added to briefly explain the indices (L.216 in the revised paper).

Expecting the estimator to also depend on the strength of seasonality, we include the two seasonality indices described in Section~\ref{section_2.3}. The SR is calculated as the ratio of the mean summer and winter low flows. Its absolute value thus indicates the strength of seasonality, with values close to one corresponding to a perfectly mixed regime and larger values pointing to stronger seasonality. For the index r, a value r=0 indicates that the low-flows are equally distributed over the year (weak seasonality) and a value r=1 that all low-flow events fall on the same day of the year (strong seasonality). It has been shown that both indices allow a good distinction of regime types and can help in identifying characteristic processes as a basis for regionalization \citep{laaha_seasonality_2006}. We also considered including a further index, the mixture rate (MR), which indicates the proportion of summer events in the annual minima series \citep{laaha_mixed_2022}. However, as the index performed somewhat worse in a preliminary evaluation, it was not included in the paper.

Concerning the question why SR is an indicator of seasonality, I refer to our earlier paper (see citation in MS above) where the skill of various seasonality indices was assessed in greater detail. Among them, the seasonality histograms (counts of low flow days per month) is the most detailed index, the circular seasonality index (mean day of occurrence and strength r) is more condensed, and the SR is the most condensed information. We found that seasonality patterns that we obtained when plotting the information for the Austrian study area are quite similar to each other, so all three indices were indeed appropriate to discern between summer, winter and mixed regimes in the study area. I agree that the index SR is more sensitive to absolute minima than durations or counts below a threshold, but all these indices are highly correlated. The proportion of summer (or winter) low flows in the annual minima series was used as a third indicator in the first paper, and termed mixture rate therein. It was used in an initial assessment but again where it led to similar results, but with somewhat weaker performance. As the SR and r index are more established we decided to show only results for SR and r here.

*Technical remarks:*

  *l. 19: use "than" instead of "as"*

  *l. 58: What does "annual drought duration of volume" mean?*

  *l. 62: I guess a "not" is missing here.*

  *l. 149: (and throughout the text) I believe it should be "Elbersee gauge" instead of "gauge Elbersee"*

  *l. 158: "of" instead of "or"?*

  *l. 163: "that the mixed probability estimator"*

  *l. 246: should be "in Table 2".*

  *l. 308: "has" instead of "have"*

  *Figure 1: The mixed Copula model should be added to the legend. Currently, it is only mentioned in the caption.*

  *Figure 3: remove "as" from the first line in the caption*

RE: Many thanks, done!

*Response to Reviewer #2*

*I would thank the reviewer for the time to review my manuscript and the constructive comments. Please find my point-by-point responses below. The line numbering in my responses refers to the revised version of the MS, unless otherwise indicated. The reviewer comments are presented in italic format, the responses in roman typeset. Additionally, to facilitate navigation through our responses I report in blue direct quotes from the MS (i.e. unchanged text) in blue color Changes (new text lines) in the revised MS are highlighted in red color.*

*1. Throughout the manuscript (See line 70 for example), the authors have claimed he is not aware of any study that used copulas to consider seasonal dependence in a mixed distribution approach. Ganguli and Reddy (2014) developed copula-based ensemble drought prediction models with up to 3 months' lead time considering the seasonality of SPI at a 6-month accumulation window. Two variants of drought forecast models were proposed: a single ensemble drought prediction model without seasonal partition and separate models for each of the four seasons in a year, combining them to constitute a yearly forecast model. The analyses showed that the seasonal prediction model performs better as compared to the model without seasonal partition. In addition, the incorporation of a copula-based conditional framework helps to provide an estimation of uncertainty.*

RE: Thanks for this citation. This study uses a copula-based model between predicted and observed SPI to provide an estimation of uncertainty of the predicted drought index. While it is an interesting application of Copula models in the field of drought indices, it appears to have a quite different objective as compared to the scope of this study. Here, we focus on the extreme value distribution, using methods of frequency analysis of extreme events, which is not the case for the cited study. Moreover, here we use Copulas to model dependencies between seasonal distributions, whereas the cited study uses copulas between marginal predicted and observed SPI to simulate uncertainty. We therefore think that our study indeed presents an innovative approach to consider seasonal dependence in a mixed distribution approach.

As said above, the study shows an interesting application of copula models and a citation has been added accordingly (L. 58): "Finally, another application of copulas can be found in Ganguli and Reddy (2014), where a copula-based model between predicted and observed drought index values was used to simulate predictive uncertainty of meteorological drought forecasts."

*2. "Eqs. 2, 6-8 representations of random variables are not correct. In fact, for summer, the random variables are drawn from the population, S = {s1, s2, ….., sn}. Likewise, for winter, the random variables are drawn from the population, W= {w1, w2, ….., wn}. The probability of a low-flow event with magnitude during summer, s and winter, w seasons to be represented as $F_S(s)$ and $F_W(w)$, respectively."…*

RE: Our formulation is based on Stedinger et al. (1993) Section 18.6.2, which formulated a specific solution of the cdf of the mixture distribution as a product $F_S(m)$ $F_W(m)$. We agree with Stedinger that this is a valid formulation for the special case when we want to estimate a low-flow event with a certain magnitude m (or q). For this purpose, we set w=q and s=q, to obtain a specific solution for the case w=s=q, which we present in our MS. This notation is not only consistent with Salinas, but also

with the first part of the study, which was already accepted for publication in this form. I like to keep this formulation and clarified this in the text as follows (L. 91):

Assuming independence of summer and winter events, the probability of a low-flow event with magnitude $q$ can be obtained from its occurrence probabilities in the summer season $F\_S(q)$ and winter season $F\_W(q)$ using the multiplication rule of statistics \citep{stedinger_frequency_1993}. By the notation F.(q) we make explicit that the same flow (i.e. the event magnitude of interest) is inserted into both marginal distributions, so that $F_S(q)=F_S(s=q)$ and $F_W(q)=F_W(w=q)$. In this case, the mixed probability estimator can be written as …

*… Further, typically subscript/sample space is shown using capital letters, whereas the argument/individual random variables are shown using small letters with the same notations. They cannot be different, for example, q as mentioned in the manuscript even after considering seasonal stratifications.*

RE: As said before, our formulation is consistent with this basic formula notation, but presented for the special case of s=w=q. We thereby follow the notation of Stedinger (see above).

*3. Line # 105, How is the copula parameter $\vartheta$ is, estimated? There is no direct relationship between Spearman's $\rho$ and $\vartheta$ for the Gumbel-Hougaard copula. However, such a relationship exists for Kendall's $\tau$ and $\vartheta$. Also, it would be good to see the gauge-wise performance of Gumbel-Hougaard copula for the summer and winter seasons either in the SI section or in the main manuscript to see how credible are the copula performance in modelling seasonal dependence.*

RE: The copula parameter is estimated base using a maximum likelihood estimator. A sentence has been added, and L. 113 was modified to make the direct reference to Kendall's $\tau$.

*4. Eq. 9, same error for random variable representation, PQ(q) instead of Pm(q).*

RE: To avoid possible confusion, we replace $p_m(q)$ by p, as the further indices are not needed here (same for Eq. 10 which is now Eq. 11).

5. *At first, Eq. 11 should appear, followed by Eq. 10. Again, representations of random variables should be corrected considering seasonal partition in line with Eq. 9 and others.*

RE: Thanks, the order of equations has been changed and the text modified accordingly. As stated above, we would like to keep the notation most similar to Stedinger's notation, in line with our earlier responses.

6. *Line #155, there is a subtle difference between the two estimators.*

The sentence has been reformulated accordingly ("there is indeed almost no difference").

7. *Line # 168, "We note that this behavior…*

RE: the text was changed accordingly

*8. Line # 173, how seasonality and seasonality ratio are determined in this study? Also, please discuss the associated implications of each of these indices in brief.*

The following text was added (L.160): Seasonality is characterized by the seasonality ratio ($SR$), where $SR > 1$ indicates a winter and $SR < 1$ a summer low flow regime, and the mean resultant of the circular seasonality index ($r$), where a value of 0 represents the weakest and a value of 1 the strongest possible seasonality. Further details about the indices are given in Section~\ref{section_3.3}.

*9. Line # 190 and paragraph therein: However, the copula-based approach is expected to preserve seasonal dependence patterns apart from the fact that they consider the marginal distribution of any form. On the other hand, the mixed distribution assumes only one type of probability distribution.*

RE: The mentioned paragraph (around line # 190 original MS) is about visual evidence. To keep the flow, I would not include theoretical properties of the estimators here.

*10. Line 195, could you provide a list of gauges, WMO ID, their latitude/longitude, catchment area, and years of available records in SI as part of the reproducibility of the work?*

RE: We are using here the dataset of Laaha et al. (2017) for demonstration, a selection of stations (mainly based on the former FRIEND archive EWA which is now embedded in the GRDC data base) which was gathered for the purpose to explore regional drought patterns across Europe. As such we do not have full station information available, mainly coordinates and flow series. Making this data set publicly available is beyond the scope of the study and I thus feel unable to provide such additional information.

*11. How the low flow is estimated in this study? Whether the constant/variable threshold approach is implemented to detect low flow signal.*

RE: The comment corresponds with comment 3 of Reviewer 1. Low flows are defined by the annual / seasonal minima series (see beginning of Section 2.1). There is no threshold selection involved. The definition of the low-flow index will be made clearer right from the beginning of the paper.

*12. Line 215: How relevant would be seasonal exceedance probability estimation since the concept of return period revolves around the sampling of annual and partial duration series?*

RE: The AMS method of frequency analysis can readily be applied for seasonal calculations, when the annual extreme are sampled from a specific season, i.e. after seasonal stratification of the complete time series. The seasonal distribution is then a characterization of the occurrence probabilities of discharges in the respective season only. It is indicative of the severity of, e.g., a summer event relative to all other summer events, which is supplemental information to the annual probability of the event. Here we do not focus on the merits of knowing the individual seasonal distributions, but on the advantages when these are recombined to an annual estimate.

*13. Eq. 13, the relative absolute deviation will not show any over/underestimation effect. Therefore, the absolute unit would be more beneficial.*

RE: I agree, and realise that the analyses were indeed performed on the relative deviation \Delta rd (and not on the relative absolute deviation), which I corrected in the revised version of the MS. As Table 5 has no added value in the given case where we have only negative deviations, it was removed.

14. Fig. 3 caption; does the uncertainty of the annual probability estimator considers all catchments across the pan-EU scale?

In principle yes, in the given range of the graphs, while a few catchments with higher values (>600 %) were excluded for the sake of visual clarity. This is now added to the figure caption: Single outliers > 600 \% are discarded.

*15. Line # 288: the large differences between two estimators at low return period is consistent with the differences in quantile estimates between the annual maxima/minima and partial duration approach. In fact, the difference in return period estimation in Annual maxima/minima vs. partial duration series is generalized using a simple exponential relationship. Can the author derive such kind of generalized formula for the given EU catchments?*

The focus of this paper is the AMS approach only. The differences between the two estimators are stemming from the fact that the annual series is not homogeneous, which violates the validity of a distribution fitted to the data. The fitted distribution can be disturbed in various ways. I therefore cannot see a direct analogy to the difference (and exponential relationship) between AMS and PDS approaches, and would not expect a generic relationship.

*16. For Figs. 2, 4-5, and 7-8, please use a continuous color bar at the bottom of the figures and show the color discretization.*

There are pros and cons between continuous and discrete colour bars. The continuous can be said to show the original information more directly, whereas the discrete color classification has the advantage that it allows the reader to read-off the value more easily. I find the discrete color better suited and would like to keep it for the study.

*17. Table 3. The minimum quantile is always zero, indicating water level is always less than the stage recorder during the low flow period; therefore, instead of furnishing information on the Min quantile, the 1st (25th percentile), median, and 3rd quantiles, including the interquartile range, would suffice. In fact, the IQR would show the catchment-wise variability in low flows for each estimator.*

RE: I have updated the table accordingly:

*18. Line # 322, VIF is not defined earlier.*

RE: The abbreviation VIF is defined in Section 3.2, where I have added a short characterization (L. 293):

The VIF provides an index that measures how much the variance of an estimated regression coefficient is increased because of collinearity. The adjusted $R^2$ is a penalized measure of model performance, so that a greater difference between adjusted and unadjusted $R^2$ will be interpreted as an indication of overfitting

*19. Line # 344: This suggests sensitivity towards BFI, which is, in turn, the function of catchment soil types and the availability of water bodies nearby.*

Yes. More general, it is an indication of discharge share from stored sources (such as groundwater, lakes, soil water, or snow storage). This was added to the text.

*20. Line 353: Laaha, 2022?*
Thanks, corrected.

*21. Line 386: the uncertainty could also stem from estimated copula parameters and the uncertainty due to marginal distribution. Therefore, sample lengths have a profound impact on multivariate distribution. A list of available sample lengths is to be presented. For credible assessment of multivariate risk, sample lengths need to be at least more than 25 or larger.*

RE: I fully agree. It is common knowledge that the sample length is an important factor of the accuracy of frequency models, and often a minimum sample length of 25 or 30 years has been recommended. Here we have 673 series with full record length of 35 years and further 80 series with a record of at least 30 years and 29 records of at least 25 years. The information has be added to the text (L. 214).

As can be seen, all stations have a common 25-year observation period, thereby satisfying both minimum record length criteria. As most stations are based on the same sample length, these sources of uncertainty can be assumed to factor out.

*22. Line 388: authors have pointed influence of catchment area, BFI, and to some extent, climate; however, terrain attributes, soil types, and land use/land cover do also have a profound impact on drought seasonality, persistence and recovery pattern.*

RE: Yes, all these characteristics contribute to low flow generation, and BFI and seasonality are good indicators of their combined effect on catchment storage and release properties of the catchment. A sentence was added accordingly (L. 447):

Storage and catchment size only have a minor effect on the performance gain of the mixed copula estimator and appear to play a subordinate role. This should also be the case for other catchment characteristics, such as soil properties, vegetation, and terrain, that tend to have more influence on surface processes and fast components of the water balance than on long-term storage, and thus less influence on the redistribution of water over time.

*23. Line 400: The pronounced differences in quantiles, mainly at low return periods, are consistent with annual vs. partial duration series, wherein the differences tend to diminish for high return period events. A generalization of this would add value.*

RE: As stated above (Comment 15), I cannot see any direct analogy with the difference between AMS and PDS approaches. The errors of the conventional annual estimator here stem from a different source (process heterogeneity of an annual extremes series).

*24. This study is useful at a regional level focusing only EU zone. A discussion on how this study would add value for other parts of the globe, for example, in monsoon-dominated regions (South Asia and Africa) where marked seasonality is pronounced or in areas with relatively stable climate with subtle climatic variability (for example, sub-tropics) would be beneficial for audiences across the globe.*

RE: I think the added value is mainly for regimes where both summer low-flows (due to precipitation deficit) and winter low-flows (due to freezing) occur, as only there the low flow processes are so fundamentally different that process-heterogeneity of annual extremes series gets thus prominent. This restricts the added value to cold and temperate climate with a cold season. I added clarified this issue as follows (L. 424):

In the subsequent step, the performance gain was evaluated based on a larger pan-European data set. The patterns match well with the findings from the exemplary catchments. There is generally little difference between the mixed distribution approaches for severe low-flow events. However, for mild low-flow events the differences are large. For the $T$=2 year event, the gains of the mixed copula approach are most pronounced in the lowlands north of the Alps which is subject to cold climate. The gains are much smaller in the west, which is subject to temperate climate. Altogether, the patterns suggest a large gain of the mixed copula estimator over the annual probability estimator, making the method highly relevant for Europe as a whole. Similar effects can be expected around the world in cold and temperate climates, where both summer low-flows (due to precipitation deficit) and winter low-flows (due to freezing) occur. The method, however, should be less relevant for other (seasonal or aseasonal) climates without a frost season, as generating processes are not so fundamentally different for the low-flow events there.

25. Last but not least, the study assesses the performance of mixed distribution vs. copula-based dependency framework for modelling low flows accounting seasonality. This message is not well reflected in the title. The title could be tuned in that direction, for example: A bivariate approach for low-flow frequency analysis considering seasonality Part 2: Comparative assessment of Mixed Probability vs Copula-based Dependence Framework.

RE: Thanks for the suggestion! The title "A mixed distribution approach for low-flow frequency analysis" was chosen on purpose, to reflect that the approach is closely related to established methods for floods, which are here transferred to low flows. These approaches were often termed "mixtures" (Stedinger et al., 1993) "mixture distribution" (Fischer and Schumann, 2021; Szulczewski and Jakubowski, 2018) or "mixed probability distribution" (Fischer et al. 2016) approaches. The main aim of both papers is to propose a mixed distribution approach for low-flow frequency analysis. Here, we extend the estimator incorporate dependency of seasonal events using a copula-based estimator. I still find the title well suited, but this may also be a matter of taste. However, as the first part is already accepted, I feel unable to modify the title in the suggested form. However, I find the suggested part 2 (Comparative assessment of Mixed Probability vs Copula-based Dependence Framework) well suited and will use it in the title.

*26. A Minor comment:*

*Line 41: Ganguli and Reddy (2012) presented a bivariate risk assessment framework for meteorological droughts in Saurashtra and Kutch regions of Gujarat state in India. Based on the tail dependence measure, the Gumbel-Hougaard copula emerged as the best model for modeling joint dependency between drought severity and duration. The comparative assessment of traditional bivariate distributions, such as bivariate log-normal and bivariate logistic models relative to copulas suggested that the extreme value family of the Gumbel-Hougaard copula was better suited for the area.*

RE: Thanks, the citation has been added as well (L. 52).

This was also confirmed in the case study of \cite{ganguli_risk_2012}, where based on the tail dependence measure the Gumbel-Hougaard copula was found to be the best suited model for capturing the joint dependence between meteorological drought severity and duration.

References:

Szulczewski, W. and Jakubowski, W.: The Application of Mixture Distribution for the Estimation of Extreme Floods in Controlled Catchment Basins, Water Resour Manage, 32, 3519–3534, https://doi.org/10.1007/s11269-018-2005-6, 2018.

Fischer, S., Schumann, A., and Schulte, M.: Characterisation of seasonal flood types according to timescales in mixed probability distributions, Journal of Hydrology, 539, 38–56, https://doi.org/10.1016/j.jhydrol.2016.05.005, 2016.

---

## Author Response (AR2)

***Author's response***

*I would like to thank the editor and the reviewers again for their valuable time spent for reading my MS. The remaining points raised by the reviewers (Table 4, some typos) have been amended.*

*Kind regards,*

*Gregor Laaha*